# Isavuconazole: Mechanism of Action, Clinical Efficacy, and Resistance

**DOI:** 10.3390/jof6040324

**Published:** 2020-11-29

**Authors:** Misti Ellsworth, Luis Ostrosky-Zeichner

**Affiliations:** 1Division of Pediatric Infectious Diseases, UT Health McGovern Medical School, Houston, TX 77030, USA; 2Division of Infectious Diseases, UT Health McGovern Medical School, Houston, TX 77030, USA; Luis.Ostrosky-Zeichner@uth.tmc.edu

**Keywords:** isavuconazole, triazole, *Aspergillus*, *Candida*, *Mucorales*, invasive fungal disease

## Abstract

Increasing incidence of invasive fungal infections combined with a growing population of immunocompromised hosts has created a rising need for antifungal agents. Isavuconazole, a second-generation broad-spectrum triazole with activity against yeasts, dimorphic fungi, and molds, has a favorable safety profile and predictable pharmacokinetics. Patients typically tolerate isavuconazole well with fewer drug–drug interactions. Clinical trials have found it to be noninferior to voriconazole for invasive aspergillosis, an alternative therapy for salvage treatment of mucormycosis, and suitable for stepdown therapy with invasive candidiasis. Cross-resistance with other triazoles is common. More studies are needed to determine the role of isavuconazole in anti-mold prophylaxis in high-risk patients.

## 1. Introduction

The incidence of invasive fungal infections (IFI) has been increasing as advances in medical treatments for cancer and other chronic health conditions create an ever-growing population of immunocompromised hosts. The mortality rate of these infections is estimated to exceed 50% [1]. The high mortality rate in combination with limited treatment options leads to over 1.5 million deaths/year worldwide [2]. Four classes of antifungal drugs comprise the arsenal to combat IFIs: azoles, echinocandins, polyenes, and flucytosine. Within the azole class, the systemic use triazoles are some of the most frequently used antifungal drugs due to improved tolerability and more favorable side effect profile when compared to polyenes [3]. One of the newest triazoles, isavuconazole, was approved by the FDA in 2015 for the treatment of invasive aspergillosis and mucormycosis. Its use has since been expanded to treat a wide array of fungal infections and as IFI prophylaxis. The aim of this review is to review the mechanism of action of isavuconazole, summarize the treatment data available, and explore mechanisms of resistance.

## 2. Mechanism of Action

Isavuconzaole inhibits cytochrome P450-dependent lanosterol 14α-demethylase, which is essential for the synthesis of ergosterol, a component of the fungal membrane. This disruption leads to alterations in the structure and function of the fungal membrane leading to cell death [4]. The isavuconazole structure includes a side arm that orients the molecule to engage the triazole ring to the binding pocket of the fungal CYP51 protein (Figure 1). This confers broader antifungal activity in comparison to other azoles [5].

## 3. Pharmacokinetics and Pharmacodynamics

Isavuconazole is the active component of isavuconazonium sulfate, a triazole precursor that is water-soluble and can be given orally or intravenously. The drug has a large volume of distribution with the majority (approx. 98%) being bound to protein [4]. Bioavailability is high, approaching 98%, and the same dose is used for oral and intravenous administration. Unlike other triazoles, isavuconazole absorption is not impacted by food intake, gastric acid-suppressing medications, or the presence of mucositis [6,7]. Previous studies have proven that isavuconazole administered in an IV or oral formulation follows dose-dependent pharmacokinetics with minimal variability among healthy subjects with the maximum plasma concentration occurring one and two to three hours after administration, respectively [8,9]. Isavuconazole is proposed to be widely distributed to the majority of tissues, including brain, liver, lung, and bone. Clinical data supporting brain penetration in humans is limited to case reports showing successful treatment of fungal CNS infections with isavuconazole [10]. Tissue distribution has been studied in rat models for both single and daily doses of isavuconazole, with findings of rapid tissue penetration [11]. The VITAL and SECURE trials showed similar pharmacokinetics with minimal intra-subject variability and narrow distributions of trough levels in patients with proven IFIs [12,13]. Similar results were found in trough levels of patients with renal failure [14]. Subsequent studies have found disparities in drug levels in patients on renal replacement therapy and extracorporeal membrane oxygenation (ECMO), and Andres et al. found therapeutic drug monitoring (TDM) may be warranted in patients who are obese, <18 years of age, or who have moderate hepatic failure [15,16,17]. Currently, neither the European Conference on Infections in Leukemia (ECIL-6) guidelines nor the Infectious Disease Society of America Guidelines for Aspergillosis recommend routine TDM for isavuconazole [18,19].

## 4. In Vitro Activity

In vitro data support broad activity for isavuconazole against yeast, molds, and dimorphic fungi. Isavuconzole is active against most *Candida* species, including *C. krusei* and *C. glabrata* [20]. In a study that compared voriconazole and isavuconazole minimum inhibitory concentrations (MIC) of 1677 *Candida* isolates found the distributions were comparable [21]. Seifert et al. evaluated 296 blood stream *Candida* isolates for in vitro activity of amphotericin B, flucytosine, fluconazole, itraconazole, voriconazole, and isavuconazole. Isavuconazole was found to have lower MICs than fluconazole and similar activity to amphotericin B, itraconazole, and voriconazole. The minimum inhibitory concentration (MIC) values for all major *Candida* species, corresponding to 50% and 90% growth inhibition, were as follows: MIC50 < 0.5 mg/L and MIC90 < 2.0 mg/L, respectively [22]. In addition to *Candida*, in vitro activity is also present for *Cryptococcus neoformans* and *Cryptococcus gattii* with MICs reported from <0.008 to 0.5 mg/L [23]. Isavuconazole activity against *C. auris* is variable, and empiric therapy should be initiated with an echinocandin until susceptibility results are available [24].

Isavuconazole plays an important role in the treatment of invasive aspergillosis. In a study evaluating 702 strains of *Aspergillus*, isavuconazole showed an MIC90 of 1 µg/mL. Isavuconazole also showed activity against *A. terreus*, a known amphotericin B resistant species [20]. In a report that evaluated isavucoazole’s in vitro activity against typically azole-resistant fungal pathogens including *A. lentulus* found isavuconazole was active despite voriconazole and itraconazole resistance [25].

A total of 72 clinical isolates of *Mucorales* were evaluated for susceptibility to amphotericin B, posaconazole, voriconazole, and isavuconazole. All isolates were susceptible to amphotericin, resistant to voriconazole, and potentially susceptible to posaconazole and isavuconazole. Of note, more isolates were found to be potentially susceptible to isavuconazole when compared to posaconazole [26].

Small studies have also shown potential activity against *Scedosporium* and *Fusarium* species; however, more in vitro data are needed before strong conclusions can be made [20].

## 5. Drug–Drug Interactions and Adverse Events

Overall, isavuconazole is well tolerated with a favorable side effect profile when compared to other agents in the azole class. The most commonly reported side effects include nausea, vomiting, and diarrhea [9,21]. Symptoms are generally not severe enough to require discontinuation of therapy. Hepatotoxicity can occur with isavuconazole and liver enzymes should be monitored while on therapy. Overall, this occurs to a lesser extent than with other commonly used triazoles. In the SECURE trial, there was a lower frequency of drug-related hepatotoxicity in the isavuconazole group versus the voriconazole group (2% vs. 10% [27]).

In contrast to other triazoles that cause prolongation of the time duration between the onset of the QRS complex and the end of the T wave, or QTc segment, isavuconazole can lead to dose-dependent QTc shortening [28]. A paper evaluating changes in the QTc interval in 26 adult patients found that 24 patients showed a mean interval decrease of 7.4 ± 5.8% [29]. With the exception of individuals with familial shortened QTc syndrome, in whom isavuconazole should be avoided, the clinical significance of QTc interval shortening remains unclear [30].

Isavuconazole is a substrate for CYP3A4, and coadministration of other medications that inhibit or induce this enzyme should be avoided. Inhibitors that can lead to increased isavuconazole levels include ketoconazole and high-dose ritonavir, and the inducers that can lead to reduced levels of isavuconazole include long-acting barbiturates, rifampin, and carbamazepine [30]. Isavuconazole is a moderate inhibitor of CYP3A4 and can lead to elevated levels of drugs such as sirolimus, tacrolimus, cyclosporine, colchine, and digoxin due to decreased metabolism. Van Kieu et al. evaluated tacrolimus/sirolimus levels in hematopoietic stem cell transplants receiving isavuconazole and found a moderate increase in the concentration/dose ratios. Serum levels of these drugs should be monitored if given concomitantly with isavuconazole [31].

## 6. Clinical Efficacy

### 6.1. Invasive Aspergillosis

Invasive aspergillosis (IA) accounts for the majority of IFIs in immunocompromised patients and results in significant mortality and morbidity. A study in hematopoietic stem cell transplant (HSCT) patients found the overall one-year survival rate to be 25.4% in patients with IA infections [32]. Treatment options for IA are limited, and the standard drugs used in treatment have significant toxicities and drug interactions. To confirm the in vitro data for isavuconazole against *Aspergillus* species, the SECURE trial was designed to compare treatment outcomes for isavuconazole versus voriconazole in patients with suspected invasive mold disease [27]. This was a Phase 3 randomized double-blind non-inferiority trial. The primary endpoint was all-cause mortality at day 42 of treatment in patients who received at least one dose of the study drug using a 10% non-inferiority margin. Secondary endpoints included all-cause mortality at day 84, overall clinical, mycological, and radiological response at day 42, 84, and the end of treatment. Safety and tolerability of both drugs was also studied. A total of 527 patients were randomized to receive either voriconazole (*n* = 264) or isavuconazole (*n* = 263) at the following doses: voriconazole 6 mg/kg IV twice daily on day one, 4 mg/kg IV twice daily on day two, followed by either IV treatment at 4 mg/kg IV twice daily or 200 mg orally twice daily or isavuconazole 200 mg IV three times daily on days one and two followed by 200 mg IV once daily IV or perorally. In each group, 258 patients received at least one dose of the study drug. Probable, proven, or possible invasive mold disease was determined using the definitions from the European Organization for Research and Treatment of Cancer/Invasive Fungal Infections (EORCT) and was confirmed by independent infectious disease experts. In total, 65 (13%) patients were determined to have proven disease, 207 (40%)—probable disease, 196 (38%)—possible disease, and 48 (9%) had no evidence of disease. The most common underlying condition were hematologic malignancies (84%), of which 66% were neutropenia; in 20% of cases, patients received allogenic hematopoietic stem cell transplants. From these patients, two modified populations were created for further analysis. The modified intention-to-treat (mITT) population was created to include ITT patients with proven or probable invasive mold disease, and a mycological intention-to-treat population (myITT) was a subset of the mITT population with proven or probable IA. When *Aspergillus* was cultured as the only mold at the baseline, *A. fumigatus*, *A. flavus*, *A. niger*, and *A. terreus* were the most commonly identified species. Efficacy analysis for treating invasive aspergillosis focused on the mITT population.

The primary endpoint of all-cause mortality at day 42 found that isavuconazole was non-inferior to voriconazole in the ITT population (18.6% for the isavuconazole arm and 20.2% for the voriconazole arm). Overall success, including complete and partial response, at the end of treatment was not different for the two drugs (35% isavuconazole, 3.4% voriconazole). The results were similar for the mITT and myITT groups for all-cause mortality (20% versus 23% and 19% versus 22%, respectively) and treatment response (35% vs. 36%, respectively). In addition, isavuconazole was associated with significantly less hepatobiliary, eye, and skin adverse events and treatment discontinuation due to adverse events was significantly lower in the isavuconazole arm.

A separate retrospective study found similar results in patients on long-term therapy for chronic pulmonary aspergillosis. Patients who received isavuconazole had significantly less adverse events when compared to patients treated with voriconazole [33].

Using economic modeling, cost-effectiveness studies have since been done in the US, the UK, and Sweden. All studies found that isavuconazole was cost-saving when compared to voriconazole. These cost savings included the cost of the drug, the cost of adverse events, and hospital readmissions [34,35,36].

Current guidelines from ECIL-6 and the European Society of Clinical Microbiology and Infectious Diseases/European Confederation of Medical Mycology recommend either voriconazole or isavuconazole as the first-line treatment for IA in patients with hematological malignancies [19,37]. Infectious Disease Society of America (IDSA) guidelines for the treatment of IA still recommend voriconazole as the first-line treatment [18].

### 6.2. Mucormycosis

Mucormycosis is a rare disease occurring primarily in patients who are immunocompromised. Prompt diagnosis and treatment is necessary due to the significantly high mortality rate of up to 90% associated with these infections. Treatment guidelines center on antifungal treatment and surgical debridement [38]. Anti-fungal agents used in treatment include amphotericin B and posaconazole. Both agents are associated with significant toxicity and adverse events. In 2015, the FDA approved the use of isavuconazole for invasive mucormycosis based on the results of the VITAL study. The VITAL study was an open-label non-comparative study of isavuconazole in adult patients with IA and renal impairment or in patients with invasive fungal disease caused by other rare fungi [39]. Within this study, 37 patients had proven (86%) or probable (14%) invasive mucormycosis as defined by the European Organization for Research and Treatment of Cancer/Mycosis Study Group definition. Patients with mucormycosis were eligible for analysis if isavuconazole was the primary treatment (defined as 4 days or less) or if they were refractory or intolerant to other antifungals. Of the 37, 21 patients had not received anti-fungal therapy, 11 had refractory disease, and 5 were intolerant of previous therapy. The most common underlying condition was hematologic malignancy (59%) with 35% having received HSCT and 27% being neutropenic. Disease involvement was pulmonary in 59% with additional site involvement in ½. Patients received an oral or IV loading regimen of isavuconazole 200 mg every 8 hours for six doses followed by 200 mg daily. Clinical efficacy of isavuconazole was also assessed using matched case–control analysis from the FungiScope: Global Emerging Fungal Infection Registry, a global database of rare invasive fungal diseases. This was in line with the FDA guidance for comparators in studies of rare diseases. In this analysis, patients with the isavuconazole primary treatment were matched up with three FungiScope patients who had received primary amphotericin B-based treatment for proven or probable mucormycosis [39].

The primary VITAL study endpoint was the overall clinical response at day 42. Secondary endpoints included assessments of overall, clinical, radiological, and mycological responses at day 42, day 84, and the end of treatment and all-cause mortality at days 42 and 84. Mortality at day 42 was 38%. When mortality was compared to patients with refractory disease or previous intolerance of prior therapy versus patients who were treated for primary mucormycosis, mortality was higher in the refractory/intolerance group (43.7% vs. 33.3%). The overall response rate at day 42 was 31.4%. End of treatment partial and compete response was 32% for primary treatment and 36% for refractory treatment. These rates were similar to those reported for amphotericin B.

The matched case–control analysis compared 21 patients from the VITAL trial who had primary treatment with isavuconazole to 33 patients treated with amphotericin from the FungiScope Registry. The all-cause mortality at day 42 and survival through day 84 did not differ between the two groups. Similar to the SECURE study, the VITAL study found isavuconazole to be well tolerated with seven patients able to extend therapy beyond 6 months [40].

Several other case reports have subsequently shown isavuconazole to be effective as the salvage therapy for mucormycosis [41,42,43].

### 6.3. Candida

Incidence of invasive candidiasis (IC) is increasing within the United States. Hospital inpatient IC is estimated to be 90/100,000 hospitalizations with mortality in patients with IC estimated to be around 40% [44,45]. Echinocandins are often used as the first-line therapy, but are limited to only IV administration. The ACTIVE trial compared IV isavuconazole to IV caspofungin followed by oral isavuconazole or voriconazole, respectively, in a Phase 3 randomized double-blind clinical trial for the primary treatment of patients with candidemia or invasive candidiasis [46]. Treatment groups received caspofungin 70 mg IV on day 1 followed by 50 mg IV daily or 200 mg of isavuconazole IV q8h on days 1–2 followed by 200 mg once daily. Both groups received IV therapy until day 10. After day ten, if neutropenia was not present, the patient was switched from IV to oral therapy at the clinician’s discretion. Doses of oral medications were as follows: voriconazole 400 mg twice daily on day 1 followed by 200 mg twice daily on all subsequent treatment days or 200 mg of oral isavuconazole once daily. Treatment continued for a minimum of 14 days after the last positive blood culture and could be extended up to 56 days. Catheter removal was recommended for all patients with candidemia. Patients were followed for 6 weeks, and clinical and laboratory assessments were performed at the baseline, days 7, 14, 28, 42, and 56, the end of IV therapy (EOIVT), and the end of therapy (EOT). All patients who received at least one dose of the study drug were included in the intention-to-treat (ITT, *n* = 440) population. Patients in the ITT population with invasive candidiasis or candidemia at the baseline were included in a modified ITT (mITT, *n* = 400) group. The primary efficacy endpoint was the overall response to therapy at the end of IV treatment (EOIVT). A successful response was defined as mycological eradication and clinical cure or improvement without the use of alternative antifungal therapy within 48 hours of the last dose of IV therapy. The secondary endpoint was the overall response to therapy at 2 weeks after the end of therapy (EOT). The mITT population was used to assess the primary and secondary endpoints. The mITT population consisted of 199 patients in the isavuconazole group and 201 in the caspofungin group. The most common *Candida* species were non-albicans. The most common species in both treatment arms were *C. albicans*, *C. tropicalis*, *C. parapsilosis*, and *C. glabrata.*

For the primary endpoint of the overall response at the EOIVT, a successful outcome occurred in 60.3% cases in the isavuconazole arm and 71.1% cases in the caspofungin arm. This did not demonstrate noninferiority of isavuconazole in comparison to caspofungin. The overall response rates 2 weeks after the EOT as well as survival on days 14 and 56 were similar in both arms.

In patients with candidemia, the clearance rate of *Candida* from the bloodstream was similar. Incidence of breakthrough or recurrent infections was slightly higher in the caspofungin group. Success rates in the patients transitioned from IV to oral therapy was 82.6% in the isavuconazole group and 77.5% in the caspofungin group. This finding supports the use of isavuconazole as a step-down therapy for candidiasis [46].

## 7. Resistance to Isavuconazole

Similar mechanisms that lead to fluconazole resistance in *Candida* species also confer resistance to isavuconazole [47]. Emergence of azole-resistant strains can occur after repeated exposure to the drugs within the azole class [48]. The mechanisms for azole resistance include overexpression of efflux pumps through the ATP binding cassette (ABC) transporter overexpression, mutations in the gene coding the target enzyme (ERG11) leading to reduction in the binding of azoles, mutations in the ERG3 gene resulting in the inability of azoles to disrupt the cell membrane [49]. Multiple mechanisms of resistance can be present in a single *Candida* strain and can lead to cross-resistance among the triazoles [47]. Sanglard et al. found that ABC transporters, specifically, CDr1 and CgCDR1, had the greatest effect on isavuconazole MICs when present in *C. albicans*. In contrast to fluconazole and voriconazole, isavuconazole MICs were less affected by major facilitator superfamily (MFS) transporters MDR1 or FLU1. Clinical failure during treatment with an azole should prompt the initiation of an alternative antifungal agent [48]. The well-described mechanism of azole resistance in *Aspergillus* species is alterations in the Cyp51A gene leading to changes in the enzyme targeted by azoles. Other potential mechanisms including efflux pumps and mutations in the promoter region of Cyp51A are less well-described [48]. The in vitro data support cross-resistance to isavuconazole in *Aspergillus* isolates with Cyp51A mutations conferring resistance to other azoles, such as voriconazole [50]. *Mucorales* species exhibit varying degrees of sensitivity to isavuconazole, and species identification and MIC testing are recommended prior to initiating therapy with this agent.

## 8. Summary

Isavuconazole has many advantages over other anti-fungal medications, including both IV and oral formulation, broad spectrum activity, predictable pharmacokinetics, and reduced adverse effects when compared to other triazoles. Isavuconazole is an excellent alternative to voriconazole for IA in patients with hematologic malignancies with significant concerns for drug interactions and toxicities. It is also a reasonable consideration for a step-down and salvage therapy for mucormycosis in patients with refractory disease and those unable to tolerate posaconazole. Although it was not found to be non-inferior to caspofungin for IC, it is an option for an oral step-down therapy when fluconazole cannot be used. Isavuconazole’s role in anti-mold prophylaxis in high-risk patients is less well-defined. Further studies are needed in this area.

## Figures and Tables

**Figure 1 jof-06-00324-f001:**
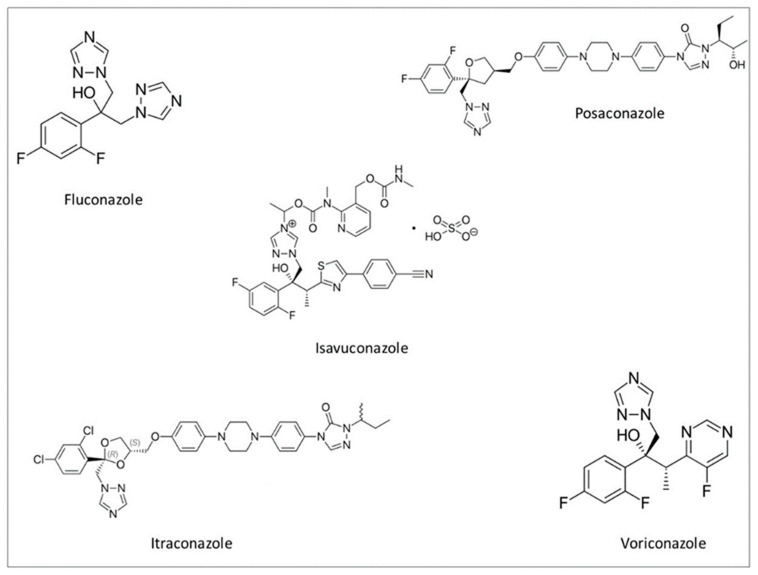
Chemical structure of fluconazole, posaconazole, isavuconazole, itraconazole, and voriconazole.

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
