# Peer review of "Isavuconazole: Mechanism of Action, Clinical Efficacy, and Resistance"

_jof, 2020, doi:10.3390/jof6040324_

Round 1
Reviewer 1 Report
The authors present a review of the mechanism of action, in vitro data and resistance mechanisms for isavuconazole. The paper provides a nice overview for JOF readers. I have a few suggestions for consideration:
- the title could be more descriptive particularly with regard to "In vitro data"; most of the data is a review of in vivo efficacy data from clinical trials. Revision of the title to better encompass the contents of the paper is suggested.
- line 35 - change "14a-sterol...." to "14α-sterol....";
- figure 1 is fuzzy and pixelated, can it be sharpened?
- lines 62, 63, 67 and 68 – Candida needs to be capitalized and italicized where it is not currently done;
- lines 69-70 – I don’t believe the statement about isavuconazole being not active against C. auris is correct; there is literature showing variable activity against C. auris. Please update and add references.
- lines 82-84 – the discussion about Zygomycetes is redundant with the paragraph above discussing the Mucorales since these are the same group of fungi (old taxonomy is Zygomycetes); please delete discussion of Zygomycetes in line 82;
- line 83 – please add “species” after “Fusarium”;
- line 236 – please replace author’s name in parentheses (Wilson) with the reference number;
- references – are double numbered and reference 23 refers to just a “®” symbol which then throws off the numbering of references after that;
Author Response
- The title could be more descriptive particularly with regard to "In vitro data"; most of the data is a review of in vivo efficacy data from clinical trials. Revision of the title to better encompass the contents of the paper is suggested.
Response: New Title: Isavuconazole – Mechanism of Action, Clinical Efficacy, and Resistance
- line 35 - change "14a-sterol...." to "14α-sterol....";
Response: Line 35 changed to: Isavuconzaole inhibits cytochrome P450 dependent a-sterol-demethylase, which is essential for ergosterol synthesis, a component of the fungal membrane.
- figure 1 is fuzzy and pixelated, can it be sharpened?
Response: The figure was adjusted to be clearer.
- lines 62, 63, 67 and 68 – Candidaneeds to be capitalized and italicized where it is not currently done;
Response: Candida has now been capitalized and italicized. See below.
In-vitro data supports broad activity for isavuconazole against yeast, molds, and dimorphic fungi. Isavuconzole is active against most Candida species including C. krusei and C. glabrata (20). In a study that compared voriconazole and isavuconazole MICs of 1677 Candida isolates found the distributions were comparable (21). Seifert et al evaluated 296 blood stream Candida isolates for in-vitro activity of amphotericin B, flucytosine, fluconazole, itraconazole, voriconazole, and isavuconazole. Isavuconazole was found to have lower MICs than fluconazole and similar activity to amphotericin B, itraconazole and voriconazole. The minimum inhibitory concentration (MIC) values for all major Candida species, corresponding to 50% and 90% growth inhibition, were (MIC50) < 0.5mg/L and (MIC90) < 2.0 mg/L respectively (22). In addition to Candida, in-vitro activity is also present for Cryptococcus neoformans and Cryptococcus gattii with MICs reported from < 0.008 – 0.5mcg/ml (23).
- lines 69-70 – I don’t believe the statement about isavuconazole being not active against aurisis correct; there is literature showing variable activity against C. auris. Please update and add references.
Response: The sentence has been changed to read: Isavuconazole activity against C. auris is variable, and empiric therapy should be initiated with an echinocandin until susceptibility results are available (24).
- lines 82-84 – the discussion about Zygomycetes is redundant with the paragraph above discussing the Mucorales since these are the same group of fungi (old taxonomy is Zygomycetes); please delete discussion of Zygomycetes in line 82;
Response: The discussion of Zygomycetes has been deleted. The sentence now reads: Small studies have also shown potential activity against Scedosporium and Fusarium species, however more in-vitro data is needed before a strong conclusion can be made (20).
- line 83 – please add “species” after “Fusarium”;
Response: Species has been added to Fusarium. Please see above response.
- line 236 – please replace author’s name in parentheses (Wilson) with the reference number;
Response: The author’s name has been replaced with the reference number: Similar mechanisms that lead to fluconazole resistance in Candida species also confer resistance to isavuconazole (47).
- references – are double numbered and reference 23 refers to just a “®” symbol which then throws off the numbering of references after that;
Response: The references have been corrected.
- Brown GD, Denning DW, Gow NA, Levitz SM, Netea MG, White TC. Hidden killers: human fungal infections. Sci Transl Med. 2012;4(165):165rv13. doi:10.1126/scitranslmed.3004404
- Bongomin F, Gago S, Oladele RO, Denning DW. Global and Multi-National Prevalence of Fungal Diseases-Estimate Precision. J Fungi (Basel). 2017;3(4):57. Published 2017 Oct 18. doi:10.3390/jof3040057
- Gallagher JC, Dodds Ashley ES, Drew RH, Perfect JR. Antifungal pharmacotherapy for invasive mould infections. Expert Opin Pharmacother. 2003;4(2):147-164. doi:10.1517/14656566.4.2.147
- Livermore J, Hope W. Evaluation of the pharmacokinetics and clinical utility of isavuconazole for treatment of invasive fungal infections. Expert Opin Drug Metab Toxicol. 2012;8(6):759-765. doi:10.1517/17425255.2012.683859
- Jenks JD, Salzer HJ, Prattes J, Krause R, Buchheidt D, Hoenigl M. Spotlight on isavuconazole in the treatment of invasive aspergillosis and mucormycosis: design, development, and place in therapy. Drug Des Devel Ther. 2018;12:1033-1044. Published 2018 Apr 30. doi:10.2147/DDDT.S145545
- Schmitt-Hoffmann A, Desai A, Kowalski D, Pearlman H, Yamazaki T, Townsend R. Isavuconazole absorption following oral administration in healthy subjects is comparable to intravenous dosing, and is not affected by food, or drugs that alter stomach pH. Int J Clin Pharmacol Ther. 2016;54(8):572-580. doi:10.5414/CP202434
- Kovanda LL, Marty FM, Maertens J, et al. Impact of Mucositis on Absorption and Systemic Drug Exposure of Isavuconazole. Antimicrob Agents Chemother. 2017;61(6):e00101-17. Published 2017 May 24. doi:10.1128/AAC.00101-17
- Schmitt-Hoffmann A, Roos B, Heep M, et al. Single-ascending-dose pharmacokinetics and safety of the novel broad-spectrum antifungal triazole BAL4815 after intravenous infusions (50, 100, and 200 milligrams) and oral administrations (100, 200, and 400 milligrams) of its prodrug, BAL8557, in healthy volunteers. Antimicrob Agents Chemother. 2006;50(1):279-285. doi:10.1128/AAC.50.1.279-285.2006
- Schmitt-Hoffmann A, Roos B, Maares J, et al. Multiple-dose pharmacokinetics and safety of the new antifungal triazole BAL4815 after intravenous infusion and oral administration of its prodrug, BAL8557, in healthy volunteers. Antimicrob Agents Chemother. 2006;50(1):286-293. doi:10.1128/AAC.50.1.286-293.2006
- Everson N, Smith J, Garner D. Successful treatment of contaminated epidural steroid associated fungal menigitis with isauvconazole [abstract]. European Congress of Clinical Microbiology and Infectious Disease (ECCMID). Copenhagen, Denmark, 2015. Abstract P0231.
- Schmitt-Hoffmann AH, Kato K, Townsend R, Potchoiba MJ, Hope WW, Andes D, Spickermann J, Schneidkraut MJ. Tissue Distribution and Elimination of Isavuconazole following Single and Repeat Oral-Dose Administration of Isavuconazonium Sulfate to Rats. Antimicrob Agents Chemother. 2017 Nov 22;61(12):e01292-17. doi: 10.1128/AAC.01292-17. PMID: 28971866; PMCID: PMC5700325.
- Kovanda LL, Desai AV, Lu Q, et al. Isavuconazole Population Pharmacokinetic Analysis Using Nonparametric Estimation in Patients with Invasive Fungal Disease (Results from the VITAL Study). Antimicrob Agents Chemother. 2016;60(8):4568-4576. Published 2016 Jul 22. doi:10.1128/AAC.00514-16
- Kaindl T, Andes D, Engelhardt M, Saulay M, Larger P, Groll AH. Variability and exposure-response relationships of isavuconazole plasma concentrations in the Phase 3 SECURE trial of patients with invasive mould diseases. J Antimicrob Chemother. 2019;74(3):761-767. doi:10.1093/jac/dky463
- Townsend RW, Akhtar S, Alcorn H, et al. Phase I trial to investigate the effect of renal impairment on isavuconazole pharmacokinetics. Eur J Clin Pharmacol. 2017;73(6):669-678. doi:10.1007/s00228-017-2213-7
- Zurl C, Waller M, Schwameis F, et al. Isavuconazole Treatment in a Mixed Patient Cohort with Invasive Fungal Infections: Outcome, Tolerability and Clinical Implications of Isavuconazole Plasma Concentrations. J Fungi (Basel). 2020;6(2):90. Published 2020 Jun 22. doi:10.3390/jof6020090
- Zhao Y, Seelhammer TG, Barreto EF, Wilson JW. Altered Pharmacokinetics and Dosing of Liposomal Amphotericin B and Isavuconazole during Extracorporeal Membrane Oxygenation. Pharmacotherapy. 2020;40(1):89-95. doi:10.1002/phar.2348
- Andes D, Kovanda L, Desai A, Kitt T, Zhao M, Walsh TJ. Isavuconazole Concentration in Real-World Practice: Consistency with Results from Clinical Trials. Antimicrob Agents Chemother. 2018;62(7):e00585-18. Published 2018 Jun 26. doi:10.1128/AAC.00585-18
- Patterson TF, Thompson GR 3rd, Denning DW, et al. Executive Summary: Practice Guidelines for the Diagnosis and Management of Aspergillosis: 2016 Update by the Infectious Diseases Society of America. Clin Infect Dis. 2016;63(4):433-442. doi:10.1093/cid/ciw444
- Tissot F, Agrawal S, Pagano L, et al. ECIL-6 guidelines for the treatment of invasive candidiasis, aspergillosis and mucormycosis in leukemia and hematopoietic stem cell transplant patients. Haematologica. 2017;102(3):433-444. doi:10.3324/haematol.2016.152900
- Guinea J, Peláez T, Recio S, Torres-Narbona M, Bouza E. In vitro antifungal activities of isavuconazole (BAL4815), voriconazole, and fluconazole against 1,007 isolates of zygomycete, Candida, Aspergillus, Fusarium, and Scedosporium species [published correction appears in Antimicrob Agents Chemother. 2008 Nov;52(11):4211]. Antimicrob Agents Chemother. 2008;52(4):1396-1400. doi:10.1128/AAC.01512-07
- Astvad KMT, Hare RK, Arendrup MC. Evaluation of the in vitro activity of isavuconazole and comparator voriconazole against 2635 contemporary clinical Candida and Aspergillus isolates. Clin Microbiol Infect. 2017;23(11):882-887. doi:10.1016/j.cmi.2017.03.023
- Seifert H, Aurbach U, Stefanik D, Cornely O. In vitro activities of isavuconazole and other antifungal agents against Candida bloodstream isolates. Antimicrob Agents Chemother. 2007;51(5):1818-1821. doi:10.1128/AAC.01217-06
- Espinel-Ingroff A, Chowdhary A, Gonzalez GM, et al. Multicenter study of isavuconazole MIC distributions and epidemiological cutoff values for the Cryptococcus neoformans-Cryptococcus gattii species complex using the CLSI M27-A3 broth microdilution method. Antimicrob Agents Chemother. 2015;59(1):666-668. doi:10.1128/AAC.04055-14
- Osei Sekyere J. Candida auris: A systematic review and meta-analysis of current updates on an emerging multidrug-resistant pathogen [published correction appears in Microbiologyopen. 2019 Aug;8(8):e00901]. Microbiologyopen. 2018;7(4):e00578. doi:10.1002/mbo3.578
- Datta K, Rhee P, Byrnes E 3rd, et al. Isavuconazole activity against Aspergillus lentulus, Neosartorya udagawae, and Cryptococcus gattii, emerging fungal pathogens with reduced azole susceptibility. J Clin Microbiol. 2013;51(9):3090-3093. doi:10.1128/JCM.01190-13
- Arendrup MC, Jensen RH, Meletiadis J. In Vitro Activity of Isavuconazole and Comparators against Clinical Isolates of the Mucorales Order. Antimicrob Agents Chemother. 2015;59(12):7735-7742. doi:10.1128/AAC.01919-15
- Keirns J, Desai A, Kowalski D, Lademacher C, Mujais S, Parker B, Schneidkraut MJ, Townsend R, Wojtkowski T, Yamazaki T, Yen M, Kowey PR. QT Interval Shortening With Isavuconazole: In Vitro and In Vivo Effects on Cardiac Repolarization. Clin Pharmacol Ther. 2017 Jun;101(6):782-790. doi: 10.1002/cpt.620. Epub 2017 Feb 13. PMID: 28074556; PMCID: PMC5485736.
- Mellinghoff SC, Bassetti M, Dörfel D, Hagel S, Lehners N, Plis A, Schalk E, Vena A, Cornely OA. Isavuconazole shortens the QTc interval. Mycoses. 2018 Apr;61(4):256-260. doi: 10.1111/myc.12731. Epub 2018 Jan 3. PMID: 29178247
- Astellas Pharma US & Inc. CRESEMBA (isavuconazonium_sulfate) prescribing information.https://www.accessdata.fda.gov/drugsatfda_docs/label/2015/207500Orig1s000lbl.pdf. Accessed September 30, 2020.
- Kieu V, Jhangiani K, Dadwal S, Nakamura R, Pon D. Effect of isavuconazole on tacrolimus and sirolimus serum concentrations in allogeneic hematopoietic stem cell transplant patients: A drug-drug interaction study. Transpl Infect Dis. 2019 Feb;21(1):e13007. doi: 10.1111/tid.13007. Epub 2018 Oct 23. PMID: 30295407.
- Kontoyiannis DP, Marr KA, Park BJ, et al. Prospective surveillance for invasive fungal infections in hematopoietic stem cell transplant recipients, 2001-2006: overview of the Transplant-Associated Infection Surveillance Network (TRANSNET) Database. Clin Infect Dis. 2010;50(8):1091-1100. doi:10.1086/651263
- Maertens JA, Raad II, Marr KA, et al. Isavuconazole versus voriconazole for primary treatment of invasive mould disease caused by Aspergillus and other filamentous fungi (SECURE): a phase 3, randomised-controlled, non-inferiority trial. Lancet. 2016;387(10020):760-769. doi:10.1016/S0140-6736(15)01159-9
- Bongomin F, Maguire N, Moore CB, Felton T, Rautemaa-Richardson R. Isavuconazole and voriconazole for the treatment of chronic pulmonary aspergillosis: A retrospective comparison of rates of adverse events. Mycoses. 2019;62(3):217-222. doi:10.1111/myc.12885
- Harrington R, Lee E, Yang H, et al. Cost-Effectiveness Analysis of Isavuconazole vs. Voriconazole as First-Line Treatment for Invasive Aspergillosis. Adv Ther. 2017;34(1):207-220. doi:10.1007/s12325-016-0443-1
- Floros L, Pagliuca A, Taie AA, et al. The cost-effectiveness of isavuconazole compared to the standard of care in the treatment of patients with invasive fungal infection prior to differential pathogen diagnosis in the United Kingdom. J Med Econ. 2020;23(1):86-97. doi:10.1080/13696998.2019.1638789
- Floros L, Kuessner D, Posthumus J, Bagshaw E, Sjölin J. Cost-effectiveness analysis of isavuconazole versus voriconazole for the treatment of patients with possible invasive aspergillosis in Sweden. BMC Infect Dis. 2019;19(1):134. Published 2019 Feb 11. doi:10.1186/s12879-019-3683-2
- Ullmann AJ, Aguado JM, Arikan-Akdagli S, et al. Diagnosis and management of Aspergillus diseases: executive summary of the 2017 ESCMID-ECMM-ERS guideline. Clin Microbiol Infect. 2018;24 Suppl 1:e1-e38. doi:10.1016/j.cmi.2018.01.002
- Petrikkos G, Skiada A, Lortholary O, Roilides E, Walsh TJ, Kontoyiannis DP. Epidemiology and clinical manifestations of mucormycosis. Clin Infect Dis. 2012;54 Suppl 1:S23-S34. doi:10.1093/cid/cir866
- Marty FM, Ostrosky-Zeichner L, Cornely OA, et al. Isavuconazole treatment for mucormycosis: a single-arm open-label trial and case-control analysis. Lancet Infect Dis. 2016;16(7):828-837. doi:10.1016/S1473-3099(16)00071-2
- Perfect JR, Cornely OA, Heep M, et al. Isavuconazole treatment for rare fungal diseases and for invasive aspergillosis in patients with renal impairment: Challenges and lessons of the VITAL trial. Mycoses. 2018;61(7):420-429. doi:10.1111/myc.12769
- Ashkenazi-Hoffnung L, Bilavsky E, Levy I, et al. Isavuconazole As Successful Salvage Therapy for Mucormycosis in Pediatric Patients. Pediatr Infect Dis J. 2020;39(8):718-724. doi:10.1097/INF.0000000000002671
- Ilharco M, Pereira CM, Moreira L, et al. Rhinoorbital mucormycosis in the immunocompetent: Experience with Isavuconazole. IDCases. 2019;18:e00591. Published 2019 Jul 9. doi:10.1016/j.idcr.2019.e00591
- Shafiq M, Ali Z, Ukani R, Brewer J. Isavuconazole: A Promising Salvage Therapy for Invasive Mucormycosis. Cureus. 2018;10(4):e2547. Published 2018 Apr 29. doi:10.7759/cureus.2547
- Ricotta EE, Lai YL, Babiker A, et al. Invasive candidiasis species distribution and trends, United States, 2009-2017 [published online ahead of print, 2020 Aug 15]. J Infect Dis. 2020;jiaa502. doi:10.1093/infdis/jiaa502
- Pfaller MA, Diekema DJ. Epidemiology of invasive candidiasis: a persistent public health problem. Clin Microbiol Rev. 2007;20(1):133-163. doi:10.1128/CMR.00029-06
- Kullberg BJ, Viscoli C, Pappas PG, Vazquez J, Ostrosky-Zeichner L, Rotstein C, Sobel JD, Herbrecht R, Rahav G, Jaruratanasirikul S, Chetchotisakd P, Van Wijngaerden E, De Waele J, Lademacher C, Engelhardt M, Kovanda L, Croos-Dabrera R, Fredericks C, Thompson GR. Isavuconazole Versus Caspofungin in the Treatment of Candidemia and Other Invasive Candida Infections: The ACTIVE Trial. Clin Infect Dis. 2019 May 30;68(12):1981-1989. doi: 10.1093/cid/ciy827. PMID: 30289478.
- Wilson DT, Dimondi VP, Johnson SW, Jones TM, Drew RH. Role of isavuconazole in the treatment of invasive fungal infections. Ther Clin Risk Manag. 2016;12:1197-1206. Published 2016 Aug 3. doi:10.2147/TCRM.S90335
- Sanglard D, Coste AT. Activity of Isavuconazole and Other Azoles against Candida Clinical Isolates and Yeast Model Systems with Known Azole Resistance Mechanisms. Antimicrob Agents Chemother. 2015;60(1):229-238. Published 2015 Oct 19. doi:10.1128/AAC.02157-15
- Pfaller MA. Antifungal drug resistance: mechanisms, epidemiology, and consequences for treatment. Am J Med. 2012 Jan;125(1 Suppl):S3-13. doi: 10.1016/j.amjmed.2011.11.001. PMID: 22196207.
- Jørgensen KM, Astvad KMT, Hare RK, Arendrup MC. EUCAST Susceptibility Testing of Isavuconazole: MIC Data for Contemporary Clinical Mold and Yeast Isolates. Antimicrob Agents Chemother. 2019 May 24;63(6):e00073-19. doi: 10.1128/AAC.00073-19. PMID: 30910898; PMCID: PMC6535523.

Reviewer 2 Report
The authors of jof-981823 manuscript aimed to provide an overview of isavuconazole mechanisms of action and resistance, PK/PD, and clinical efficacy. In general, the manuscript is well written. However, the quality/comprehensiveness of the information varies between the chapters. See the detailed comments below.
MAJOR COMMENTS
1. Chapters 2 and 3 - I strongly suggest reorganizing these chapters.
- Chapter 3: The authors do not provide information on isavuconazole pharmacodynamics but only on its spectrum of activity.
- Separation of PK and PD makes it confusing. Instead, one chapter could contain the mechanism of action + spectrum of in vitro activity, and the other one could describe PK/PD.
- The information on PK/PD is scarce and should be substantially improved/expanded (animal and human PK/PD).
2. Chapter 5 - the strongest part of the manuscript -> this chapter provides the most comprehensive information.
3. Chapter 6 - Description of resistance mechanisms is too superficial. Each resistance type (mutations in ERG11, overexpression of ERG11, overexpression of drug efflux pumps, etc.) should be explained and more information on its presence in different genera/species added. For example, if you have an isolate of Candida albicans or Candida glabrata - what mechanisms of resistance should you screen for in these two different species?
4. References: all references have double numbering. Moreover, one reference, at position 22 (prescribing information from Astellas website), is given additional numbers (23 and 24) on the list.
References are associated with information incorrectly. E.g. Lines 113-115 contain information about SECURE trial and are associated with Ref # 25 which is not on this subject; Lines 161- 164 contain information about VITAL study and are associated with Ref # 32 which is not on this subject, etc.
MINOR COMMENTS
- Introduction: isavuconazole should be written without a capital letter.
- Lines 62, 63, 67, 68: should be Candida, not candida
- Lines 66-68: sentence doesn't read well. Should rather be sth like: The minimum inhibitory concentration (MIC) values for all major Candida species, reflecting 50% and 90% growth inhibition, were <0.5 mg/L (MIC50) and <2.0 mg/L (MIC90).
- Line 72: correct unit - should be mg/L or μg/mL
- Line 73: should be species not strain
- Lines 157-161: references needed.
- Lines 174-179: references needed.
- Lines 199-229: References needed.
- Lines 233, 236: gene names (ERG11, ERG3) should be written in italics
Author Response
Major comments
Comment 1:
Chapters 2 and 3 - I strongly suggest reorganizing these chapters.
- Chapter 3: The authors do not provide information on isavuconazole pharmacodynamics but only on its spectrum of activity.
- Separation of PK and PD makes it confusing. Instead, one chapter could contain the mechanism of action + spectrum of in vitro activity, and the other one could describe PK/PD.
- The information on PK/PD is scarce and should be substantially improved/expanded (animal and human PK/PD).
Response: The paper now combines PK/PD. PK/PD information includes animal and human data. Please see below:
- PHARMACOKINETICS AND PHARMACODYNAMICS
Isavuconazole is the active component of isavuconazonium sulfate, a triazole precursor that is water soluble and can be given orally or intravenously. The drug has a large volume of distribution with the majority (approx. 98%) being bound to protein (4). Bioavailability is high, approaching 98%, and the same dose is used for oral and intravenous administration. Unlike other triazoles, isavuconazole absorption is not impacted by food intake, gastric acid suppressing medications, or the presence of mucositis (6,7). Previous studies have proven that isavuconazole administered in IV or oral formulation follows dose-dependent pharmacokinetics with minimal variability among healthy subjects with the maximum plasma concentration occurring one and two to three hours after administration respectively (8,9). Isavuconazole is proposed to be widely distributed to the majority of tissues including brain, liver, lung and bone. Clinical data supporting brain penetration in humans is limited to case reports showing successful treatment of fungal CNS infections with isavuconazole (10). Tissue distribution has been studied in rat animal models for both single and daily doses of isavuconazole, with findings of rapid tissue penetration (11). The VITAL and SECURE trials showed similar pharmacokinetics with minimal intra subject variability and narrow distributions of trough levels in patients with proven IFIs (12,13). Similar results were found in trough levels of patients with renal failure (14). Subsequent studies have found disparities in drug levels in patients on renal replacement therapy and ECMO, and Andres et al found therapeutic drug monitoring (TDM) may be warranted in patients who are obese, < 18 years of age, or who have moderate hepatic failure (15,16,17). Currently neither the European Conference on Infections in Leukemia (ECIL-6) guidelines nor the Infectious Disease Society of America Guidelines for Aspergillosis recommend routine TDM for isavuconazole (18, 19).
Comments 2 and 3:
Chapter 5 - the strongest part of the manuscript -> this chapter provides the most comprehensive information.
Chapter 6 - Description of resistance mechanisms is too superficial. Each resistance type (mutations in ERG11, overexpression of ERG11, overexpression of drug efflux pumps, etc.) should be explained and more information on its presence in different genera/species added. For example, if you have an isolate of Candida albicansor Candida glabrata - what mechanisms of resistance should you screen for in these two different species?
Response: Details are added to the resistance mechanisms that are clinically important to isavuconazole. Please see below for revised section:
- RESISTANCE TO ISAVUCONAZOLE
Similar mechanisms that lead to fluconazole resistance in Candida species also confer resistance to isavuconazole (47). Emergence of azole resistant strains can occur after repeated exposures drugs within the azole class (48). Mechanisms for azole resistance include over expression of efflux pumps through ATP binding cassette (ABC) transporter overexpression, mutations in the gene coding the target enzyme (ERG11) leading to reduction in binding of azoles, mutations in the ERG3 gene resulting in the inability of azoles to disrupt the cell membrane (49). Multiple mechanisms of resistance can be present in a single Candida strain and can lead to cross-resistance among the triazoles (47). Sanglard et al found that ABC transporters, specifically CDr1 and CgCDR1, had the greatest effect on isavuconazole MICs. In contrast to fluconazole and voriconazole, isavuconazole MICs were less effected by MFS transporters MDR1 or FLU1. Clinical failure during treatment with an azole should prompt the initiation of an alternative antifungal agent (48). The most well described mechanism of azole resistance in Aspergillus species is alterations in the Cyp51A gene leading to changes in the enzyme targeted by azoles. Other potential mechanisms including efflux pumps and mutations in the promoter region of Cyp51A are less well described (48). In vitro data supports cross-resistance to isavuconazole in Aspergillus isolates with Cyp51A mutations conferring resistance to other azoles such as voriconazole (49). Mucorales species exhibit varying degrees of sensitivity to isavuconazole, and species identification and MIC testing is recommended prior to initiating therapy with this agent.
Comment 4:
References: all references have double numbering. Moreover, one reference, at position 22 (prescribing information from Astellas website), is given additional numbers (23 and 24) on the list.
References are associated with information incorrectly. E.g. Lines 113-115 contain information about SECURE trial and are associated with Ref # 25 which is not on this subject; Lines 161- 164 contain information about VITAL study and are associated with Ref # 32 which is not on this subject, etc.
Response: The references are now correctly numbered and the double numbering is resolved. Please see below:
References
- Brown GD, Denning DW, Gow NA, Levitz SM, Netea MG, White TC. Hidden killers: human fungal infections. Sci Transl Med. 2012;4(165):165rv13. doi:10.1126/scitranslmed.3004404
- Bongomin F, Gago S, Oladele RO, Denning DW. Global and Multi-National Prevalence of Fungal Diseases-Estimate Precision. J Fungi (Basel). 2017;3(4):57. Published 2017 Oct 18. doi:10.3390/jof3040057
- Gallagher JC, Dodds Ashley ES, Drew RH, Perfect JR. Antifungal pharmacotherapy for invasive mould infections. Expert Opin Pharmacother. 2003;4(2):147-164. doi:10.1517/14656566.4.2.147
- Livermore J, Hope W. Evaluation of the pharmacokinetics and clinical utility of isavuconazole for treatment of invasive fungal infections. Expert Opin Drug Metab Toxicol. 2012;8(6):759-765. doi:10.1517/17425255.2012.683859
- Jenks JD, Salzer HJ, Prattes J, Krause R, Buchheidt D, Hoenigl M. Spotlight on isavuconazole in the treatment of invasive aspergillosis and mucormycosis: design, development, and place in therapy. Drug Des Devel Ther. 2018;12:1033-1044. Published 2018 Apr 30. doi:10.2147/DDDT.S145545
- Schmitt-Hoffmann A, Desai A, Kowalski D, Pearlman H, Yamazaki T, Townsend R. Isavuconazole absorption following oral administration in healthy subjects is comparable to intravenous dosing, and is not affected by food, or drugs that alter stomach pH. Int J Clin Pharmacol Ther. 2016;54(8):572-580. doi:10.5414/CP202434
- Kovanda LL, Marty FM, Maertens J, et al. Impact of Mucositis on Absorption and Systemic Drug Exposure of Isavuconazole. Antimicrob Agents Chemother. 2017;61(6):e00101-17. Published 2017 May 24. doi:10.1128/AAC.00101-17
- Schmitt-Hoffmann A, Roos B, Heep M, et al. Single-ascending-dose pharmacokinetics and safety of the novel broad-spectrum antifungal triazole BAL4815 after intravenous infusions (50, 100, and 200 milligrams) and oral administrations (100, 200, and 400 milligrams) of its prodrug, BAL8557, in healthy volunteers. Antimicrob Agents Chemother. 2006;50(1):279-285. doi:10.1128/AAC.50.1.279-285.2006
- Schmitt-Hoffmann A, Roos B, Maares J, et al. Multiple-dose pharmacokinetics and safety of the new antifungal triazole BAL4815 after intravenous infusion and oral administration of its prodrug, BAL8557, in healthy volunteers. Antimicrob Agents Chemother. 2006;50(1):286-293. doi:10.1128/AAC.50.1.286-293.2006
- Everson N, Smith J, Garner D. Successful treatment of contaminated epidural steroid associated fungal menigitis with isauvconazole [abstract]. European Congress of Clinical Microbiology and Infectious Disease (ECCMID). Copenhagen, Denmark, 2015. Abstract P0231.
- Schmitt-Hoffmann AH, Kato K, Townsend R, Potchoiba MJ, Hope WW, Andes D, Spickermann J, Schneidkraut MJ. Tissue Distribution and Elimination of Isavuconazole following Single and Repeat Oral-Dose Administration of Isavuconazonium Sulfate to Rats. Antimicrob Agents Chemother. 2017 Nov 22;61(12):e01292-17. doi: 10.1128/AAC.01292-17. PMID: 28971866; PMCID: PMC5700325.
- Kovanda LL, Desai AV, Lu Q, et al. Isavuconazole Population Pharmacokinetic Analysis Using Nonparametric Estimation in Patients with Invasive Fungal Disease (Results from the VITAL Study). Antimicrob Agents Chemother. 2016;60(8):4568-4576. Published 2016 Jul 22. doi:10.1128/AAC.00514-16
- Kaindl T, Andes D, Engelhardt M, Saulay M, Larger P, Groll AH. Variability and exposure-response relationships of isavuconazole plasma concentrations in the Phase 3 SECURE trial of patients with invasive mould diseases. J Antimicrob Chemother. 2019;74(3):761-767. doi:10.1093/jac/dky463
- Townsend RW, Akhtar S, Alcorn H, et al. Phase I trial to investigate the effect of renal impairment on isavuconazole pharmacokinetics. Eur J Clin Pharmacol. 2017;73(6):669-678. doi:10.1007/s00228-017-2213-7
- Zurl C, Waller M, Schwameis F, et al. Isavuconazole Treatment in a Mixed Patient Cohort with Invasive Fungal Infections: Outcome, Tolerability and Clinical Implications of Isavuconazole Plasma Concentrations. J Fungi (Basel). 2020;6(2):90. Published 2020 Jun 22. doi:10.3390/jof6020090
- Zhao Y, Seelhammer TG, Barreto EF, Wilson JW. Altered Pharmacokinetics and Dosing of Liposomal Amphotericin B and Isavuconazole during Extracorporeal Membrane Oxygenation. Pharmacotherapy. 2020;40(1):89-95. doi:10.1002/phar.2348
- Andes D, Kovanda L, Desai A, Kitt T, Zhao M, Walsh TJ. Isavuconazole Concentration in Real-World Practice: Consistency with Results from Clinical Trials. Antimicrob Agents Chemother. 2018;62(7):e00585-18. Published 2018 Jun 26. doi:10.1128/AAC.00585-18
- Patterson TF, Thompson GR 3rd, Denning DW, et al. Executive Summary: Practice Guidelines for the Diagnosis and Management of Aspergillosis: 2016 Update by the Infectious Diseases Society of America. Clin Infect Dis. 2016;63(4):433-442. doi:10.1093/cid/ciw444
- Tissot F, Agrawal S, Pagano L, et al. ECIL-6 guidelines for the treatment of invasive candidiasis, aspergillosis and mucormycosis in leukemia and hematopoietic stem cell transplant patients. Haematologica. 2017;102(3):433-444. doi:10.3324/haematol.2016.152900
- Guinea J, Peláez T, Recio S, Torres-Narbona M, Bouza E. In vitro antifungal activities of isavuconazole (BAL4815), voriconazole, and fluconazole against 1,007 isolates of zygomycete, Candida, Aspergillus, Fusarium, and Scedosporium species [published correction appears in Antimicrob Agents Chemother. 2008 Nov;52(11):4211]. Antimicrob Agents Chemother. 2008;52(4):1396-1400. doi:10.1128/AAC.01512-07
- Astvad KMT, Hare RK, Arendrup MC. Evaluation of the in vitro activity of isavuconazole and comparator voriconazole against 2635 contemporary clinical Candida and Aspergillus isolates. Clin Microbiol Infect. 2017;23(11):882-887. doi:10.1016/j.cmi.2017.03.023
- Seifert H, Aurbach U, Stefanik D, Cornely O. In vitro activities of isavuconazole and other antifungal agents against Candida bloodstream isolates. Antimicrob Agents Chemother. 2007;51(5):1818-1821. doi:10.1128/AAC.01217-06
- Espinel-Ingroff A, Chowdhary A, Gonzalez GM, et al. Multicenter study of isavuconazole MIC distributions and epidemiological cutoff values for the Cryptococcus neoformans-Cryptococcus gattii species complex using the CLSI M27-A3 broth microdilution method. Antimicrob Agents Chemother. 2015;59(1):666-668. doi:10.1128/AAC.04055-14
- Osei Sekyere J. Candida auris: A systematic review and meta-analysis of current updates on an emerging multidrug-resistant pathogen [published correction appears in Microbiologyopen. 2019 Aug;8(8):e00901]. Microbiologyopen. 2018;7(4):e00578. doi:10.1002/mbo3.578
- Datta K, Rhee P, Byrnes E 3rd, et al. Isavuconazole activity against Aspergillus lentulus, Neosartorya udagawae, and Cryptococcus gattii, emerging fungal pathogens with reduced azole susceptibility. J Clin Microbiol. 2013;51(9):3090-3093. doi:10.1128/JCM.01190-13
- Arendrup MC, Jensen RH, Meletiadis J. In Vitro Activity of Isavuconazole and Comparators against Clinical Isolates of the Mucorales Order. Antimicrob Agents Chemother. 2015;59(12):7735-7742. doi:10.1128/AAC.01919-15
- Keirns J, Desai A, Kowalski D, Lademacher C, Mujais S, Parker B, Schneidkraut MJ, Townsend R, Wojtkowski T, Yamazaki T, Yen M, Kowey PR. QT Interval Shortening With Isavuconazole: In Vitro and In Vivo Effects on Cardiac Repolarization. Clin Pharmacol Ther. 2017 Jun;101(6):782-790. doi: 10.1002/cpt.620. Epub 2017 Feb 13. PMID: 28074556; PMCID: PMC5485736.
- Mellinghoff SC, Bassetti M, Dörfel D, Hagel S, Lehners N, Plis A, Schalk E, Vena A, Cornely OA. Isavuconazole shortens the QTc interval. Mycoses. 2018 Apr;61(4):256-260. doi: 10.1111/myc.12731. Epub 2018 Jan 3. PMID: 29178247
- Astellas Pharma US & Inc. CRESEMBA (isavuconazonium_sulfate) prescribing information.https://www.accessdata.fda.gov/drugsatfda_docs/label/2015/207500Orig1s000lbl.pdf. Accessed September 30, 2020.
- Kieu V, Jhangiani K, Dadwal S, Nakamura R, Pon D. Effect of isavuconazole on tacrolimus and sirolimus serum concentrations in allogeneic hematopoietic stem cell transplant patients: A drug-drug interaction study. Transpl Infect Dis. 2019 Feb;21(1):e13007. doi: 10.1111/tid.13007. Epub 2018 Oct 23. PMID: 30295407.
- Kontoyiannis DP, Marr KA, Park BJ, et al. Prospective surveillance for invasive fungal infections in hematopoietic stem cell transplant recipients, 2001-2006: overview of the Transplant-Associated Infection Surveillance Network (TRANSNET) Database. Clin Infect Dis. 2010;50(8):1091-1100. doi:10.1086/651263
- Maertens JA, Raad II, Marr KA, et al. Isavuconazole versus voriconazole for primary treatment of invasive mould disease caused by Aspergillus and other filamentous fungi (SECURE): a phase 3, randomised-controlled, non-inferiority trial. Lancet. 2016;387(10020):760-769. doi:10.1016/S0140-6736(15)01159-9
- Bongomin F, Maguire N, Moore CB, Felton T, Rautemaa-Richardson R. Isavuconazole and voriconazole for the treatment of chronic pulmonary aspergillosis: A retrospective comparison of rates of adverse events. Mycoses. 2019;62(3):217-222. doi:10.1111/myc.12885
- Harrington R, Lee E, Yang H, et al. Cost-Effectiveness Analysis of Isavuconazole vs. Voriconazole as First-Line Treatment for Invasive Aspergillosis. Adv Ther. 2017;34(1):207-220. doi:10.1007/s12325-016-0443-1
- Floros L, Pagliuca A, Taie AA, et al. The cost-effectiveness of isavuconazole compared to the standard of care in the treatment of patients with invasive fungal infection prior to differential pathogen diagnosis in the United Kingdom. J Med Econ. 2020;23(1):86-97. doi:10.1080/13696998.2019.1638789
- Floros L, Kuessner D, Posthumus J, Bagshaw E, Sjölin J. Cost-effectiveness analysis of isavuconazole versus voriconazole for the treatment of patients with possible invasive aspergillosis in Sweden. BMC Infect Dis. 2019;19(1):134. Published 2019 Feb 11. doi:10.1186/s12879-019-3683-2
- Ullmann AJ, Aguado JM, Arikan-Akdagli S, et al. Diagnosis and management of Aspergillus diseases: executive summary of the 2017 ESCMID-ECMM-ERS guideline. Clin Microbiol Infect. 2018;24 Suppl 1:e1-e38. doi:10.1016/j.cmi.2018.01.002
- Petrikkos G, Skiada A, Lortholary O, Roilides E, Walsh TJ, Kontoyiannis DP. Epidemiology and clinical manifestations of mucormycosis. Clin Infect Dis. 2012;54 Suppl 1:S23-S34. doi:10.1093/cid/cir866
- Marty FM, Ostrosky-Zeichner L, Cornely OA, et al. Isavuconazole treatment for mucormycosis: a single-arm open-label trial and case-control analysis. Lancet Infect Dis. 2016;16(7):828-837. doi:10.1016/S1473-3099(16)00071-2
- Perfect JR, Cornely OA, Heep M, et al. Isavuconazole treatment for rare fungal diseases and for invasive aspergillosis in patients with renal impairment: Challenges and lessons of the VITAL trial. Mycoses. 2018;61(7):420-429. doi:10.1111/myc.12769
- Ashkenazi-Hoffnung L, Bilavsky E, Levy I, et al. Isavuconazole As Successful Salvage Therapy for Mucormycosis in Pediatric Patients. Pediatr Infect Dis J. 2020;39(8):718-724. doi:10.1097/INF.0000000000002671
- Ilharco M, Pereira CM, Moreira L, et al. Rhinoorbital mucormycosis in the immunocompetent: Experience with Isavuconazole. IDCases. 2019;18:e00591. Published 2019 Jul 9. doi:10.1016/j.idcr.2019.e00591
- Shafiq M, Ali Z, Ukani R, Brewer J. Isavuconazole: A Promising Salvage Therapy for Invasive Mucormycosis. Cureus. 2018;10(4):e2547. Published 2018 Apr 29. doi:10.7759/cureus.2547
- Ricotta EE, Lai YL, Babiker A, et al. Invasive candidiasis species distribution and trends, United States, 2009-2017 [published online ahead of print, 2020 Aug 15]. J Infect Dis. 2020;jiaa502. doi:10.1093/infdis/jiaa502
- Pfaller MA, Diekema DJ. Epidemiology of invasive candidiasis: a persistent public health problem. Clin Microbiol Rev. 2007;20(1):133-163. doi:10.1128/CMR.00029-06
- Kullberg BJ, Viscoli C, Pappas PG, Vazquez J, Ostrosky-Zeichner L, Rotstein C, Sobel JD, Herbrecht R, Rahav G, Jaruratanasirikul S, Chetchotisakd P, Van Wijngaerden E, De Waele J, Lademacher C, Engelhardt M, Kovanda L, Croos-Dabrera R, Fredericks C, Thompson GR. Isavuconazole Versus Caspofungin in the Treatment of Candidemia and Other Invasive Candida Infections: The ACTIVE Trial. Clin Infect Dis. 2019 May 30;68(12):1981-1989. doi: 10.1093/cid/ciy827. PMID: 30289478.
- Wilson DT, Dimondi VP, Johnson SW, Jones TM, Drew RH. Role of isavuconazole in the treatment of invasive fungal infections. Ther Clin Risk Manag. 2016;12:1197-1206. Published 2016 Aug 3. doi:10.2147/TCRM.S90335
- Sanglard D, Coste AT. Activity of Isavuconazole and Other Azoles against Candida Clinical Isolates and Yeast Model Systems with Known Azole Resistance Mechanisms. Antimicrob Agents Chemother. 2015;60(1):229-238. Published 2015 Oct 19. doi:10.1128/AAC.02157-15
- Pfaller MA. Antifungal drug resistance: mechanisms, epidemiology, and consequences for treatment. Am J Med. 2012 Jan;125(1 Suppl):S3-13. doi: 10.1016/j.amjmed.2011.11.001. PMID: 22196207.
- Jørgensen KM, Astvad KMT, Hare RK, Arendrup MC. EUCAST Susceptibility Testing of Isavuconazole: MIC Data for Contemporary Clinical Mold and Yeast Isolates. Antimicrob Agents Chemother. 2019 May 24;63(6):e00073-19. doi: 10.1128/AAC.00073-19. PMID: 30910898; PMCID: PMC6535523.
Minor comments
Comment 1:
Introduction: isavuconazole should be written without a capital letter.
Response: Isavuconazole is now written without a capital letter. See below.
One of the newest triazoles, isavuconazole, was approved by the FDA in 2015 for the treatment of invasive aspergillosis and mucormycosis. Its use has since been expanded to treat a wide array of fungal infections and as IFI prophylaxis. The aim of this review is to review the mechanism of action of isavuconazole, summarize the treatment data available, and explore mechanisms of resistance.
Comment 2:
Lines 62, 63, 67, 68: should be Candida, not candida
Response: Candida is now capitalized and italicized. See below:
In-vitro data supports broad activity for isavuconazole against yeast, molds, and dimorphic fungi. Isavuconzole is active against most Candida species including C. krusei and C. glabrata (20). In a study that compared voriconazole and isavuconazole MICs of 1677 Candida isolates found the distributions were comparable (21). Seifert et al evaluated 296 blood stream Candida isolates for in-vitro activity of amphotericin B, flucytosine, fluconazole, itraconazole, voriconazole, and isavuconazole.
Comment 3:
Lines 66-68: sentence doesn't read well. Should rather be sth like: The minimum inhibitory concentration (MIC) values for all major Candida species, reflecting 50% and 90% growth inhibition, were <0.5 mg/L (MIC50) and <2.0 mg/L (MIC90).
Response: The sentence is now restructured. Please see below.
The minimum inhibitory concentration (MIC) values for all major Candida species, corresponding to 50% and 90% growth inhibition, were (MIC50) < 0.5mg/L and (MIC90) < 2.0 mg/L respectively (22).
Comments 4 and 5:
Line 72: correct unit - should be mg/L or μg/mL
Line 73: should be species not strain
Response: The unit is now correct and strain has been replaced with species.
Isavuconazole plays an important role in the treatment of invasive aspergillosis. In a study evaluating 702 species of Aspergillus, isavuconazole showed an MIC90 of 1g/ml.
Comment 6:
Lines 157-161: references needed
Response: Reference added. Please see below.
Mucormycosis is a rare disease occurring primarily in patients who are immunocompromised. Prompt diagnosis and treatment is necessary due to the significantly high mortality rate of up to 90% associated with these infections. Treatment guidelines center on antifungal treatment and surgical debridement (38).
Comment 7:
Lines 174-179: references needed
Response: References added. Please see below.
Clinical efficacy of isavuconazole was also assessed using matched case-control analysis from FungiScope: Global Emerging Fungal Infection Registry, a global database of rare invasive fungal diseases. This was in line with FDA guidance for comparators in studies of rare diseases. In this analysis, patients with isavuconazole primary treatment were matched up with three FungiScope patients who had received primary amphotericin B based treatment for proven or probable mucormycosis (39).
Comment 8:
Lines 199-229: References needed
Response: References added. Please see below.
The ACTIVE trial compared IV isavuconazole to IV caspofungin followed by oral isavuconazole or voriconazole respectively in a Phase 3, randomized, double-blind clinical trial for the primary treatment of patients with candidemia or invasive candidiasis (46). Treatment groups received caspofungin 750mg IV day 1 followed by 50mg IV daily or 200mg isavuconazole IV Q 8 day 1-2, followed by 200mg once daily. Both groups received IV therapy until day 10. After day ten, if neutropenia was not present, the patient was switched from IV to oral therapy at the clinician’s discretion. Doses of oral medications were voriconazole 400mg twice daily day 1 followed by 200mg twice daily all subsequent treatment days or 200mg oral isavuconazole once daily. Treatment was continued for a minimum of 14 days after the last positive blood culture and could be extended up to 56 days. Catheter removal was recommended for all patients with candidemia. Patients were followed for 6 weeks, and clinical and laboratory assessments were performed at baseline, days 7, 14, 28, 42 and 56, the end of IV therapy (EOIVT), and the end of therapy (EOT). All patients who received at least one dose of study drug were included in the intention to treat (ITT n=440) population. Patients in the ITT population with invasive candidiasis or candidemia at baseline were included in a modified ITT (mITT n=400). The primary efficacy endpoint was overall response to therapy at the end of TV treatment (EOIVT). A successful response was defined as mycological eradication and clinical cure or improvement without the use of alternative antifungal therapy within 48 hours of the last dose of IV therapy. The secondary endpoint was the overall response to therapy at 2 weeks after the end of therapy (EOT). The mITT population was used to access the primary and secondary endpoints. The mITT population consisted of 199 patients in the isavuconazole group and 201 in the caspofungin group. The most common Candida species were non-albicans. The most common species in both treatment arms were C. albicans, C. tropicalis, C. parapsilosis, and C. glabrata.
For the primary endpoint of overall response at EOIVT, a successful outcome occurred in 60.3% of the isavuconazole arm and 71.1% in the caspofungin arm. This did not demonstrate noninferiority of isavuconazole in comparison to caspofungin. Overall response rates 2 weeks after EOT as well as survival on day 14 and 56 were similar in both arms.
In patients with candidemia, the clearance rate of Candida from the bloodstream was similar. Incidence of breakthrough or recurrent infections was slightly higher in the caspofungin group. Success rates in the patients transitioned from IV to oral therapy was 82.6% in the isavuconazole group and 77.5% in the caspofugin group. This finding supports the use of isavuconazole as step-down therapy for candidiasis (46).
Comment 9:
Lines 233, 236: gene names (ERG11, ERG3) should be written in italics
Response: Gene names now italicized.
Mechanisms for azole resistance include over expression of efflux pumps through ATP binding cassette (ABC) transporter overexpression, mutations in the gene coding the target enzyme (ERG11) leading to reduction in binding of azoles, mutations in the ERG3 gene resulting in the inability of azoles to disrupt the cell membrane (49).
Round 2
Reviewer 2 Report
The authors responded well to the comments from the first round of reviews.
However, there are still a few minor corrections to be made:
- Line 35: should be "lanosterol 14α-demethylase" not "14α-sterol-demethylase.
- Line 78: should be mg/L.
- Line 81: should be strains, not species.
- Line 82: should be species, not strain.
- Lines 245-247: Which Candida species is this information referring to?
- Whole manuscript: data is a plural noun and should be accompanied by plural verbs.
Author Response
- Line 35: should be "lanosterol 14α-demethylase" not "14α-sterol-demethylase.
Response: Sentence has been changed to reflect recommendations: Isavuconzaole inhibits cytochrome P450 dependent lanosterol 14a-demethylase, which is essential for ergosterol synthesis, a component of the fungal membrane.
- Line 78: should be mg/L.
Response: Units have been corrected as recommended: In addition to Candida, in-vitro activity is also present for Cryptococcus neoformans and Cryptococcus gattii with MICs reported from < 0.008 – 0.5mg/L.
- Line 81: should be strains, not species.
Response: The word species has been replaced with strains: Isavuconazole plays an important role in the treatment of invasive aspergillosis. In a study evaluating 702 strains of Aspergillus, isavuconazole showed an MIC90 of 1mg/ml.
- Line 82: should be species, not strain.
Response: The word species has been replaced by strain: Isavuconazole also showed activity against A. terreus, a known amphotericin B resistant species (20).
- Lines 245-247: Which Candidaspecies is this information referring to?
Response: The species has been added: Sanglard et al found that ABC transporters, specifically CDr1 and CgCDR1, had the greatest effect on isavuconazole MICs when present in albicans.
- Whole manuscript: data is a plural noun and should be accompanied by plural verbs.
Response: Verbs corrected after the word data used in the manuscript.